# Melatonin as a Chronobiotic/Cytoprotective Agent in REM Sleep Behavior Disorder

**DOI:** 10.3390/brainsci13050797

**Published:** 2023-05-13

**Authors:** Daniel P. Cardinali, Arturo Garay

**Affiliations:** 1CENECON, Faculty of Medical Sciences, University of Buenos Aires, Buenos Aires C1431FWO, Argentina; 2Unidad de Medicina del Sueño-Sección Neurología, Centro de Educación Médica e Investigaciones Clínicas “Norberto Quirno” (CEMIC), Buenos Aires C1431FWO, Argentina; adcgaray@gmail.com

**Keywords:** α-synucleinopathies, circadian rhythms, melatonin, neurodegeneration, oxidative stress, sleep, parasomnias

## Abstract

Dream-enactment behavior that emerges during episodes of rapid eye movement (REM) sleep without muscle atonia is a parasomnia known as REM sleep behavior disorder (RBD). RBD constitutes a prodromal marker of α-synucleinopathies and serves as one of the best biomarkers available to predict diseases such as Parkinson disease, multiple system atrophy and dementia with Lewy bodies. Most patients showing RBD will convert to an α-synucleinopathy about 10 years after diagnosis. The diagnostic advantage of RBD relies on the prolonged prodromal time, its predictive power and the absence of disease-related treatments that could act as confounders. Therefore, patients with RBD are candidates for neuroprotection trials that delay or prevent conversion to a pathology with abnormal α-synuclein metabolism. The administration of melatonin in doses exhibiting a chronobiotic/hypnotic effect (less than 10 mg daily) is commonly used as a first line treatment (together with clonazepam) of RBD. At a higher dose, melatonin may also be an effective cytoprotector to halt α-synucleinopathy progression. However, allometric conversion doses derived from animal studies (in the 100 mg/day range) are rarely employed clinically regardless of the demonstrated absence of toxicity of melatonin in phase 1 pharmacological studies with doses up to 100 mg in normal volunteers. This review discusses the application of melatonin in RBD: (a) as a symptomatic treatment in RBD; (b) as a possible disease-modifying treatment in α-synucleinopathies. To what degree melatonin has therapeutic efficacy in the prevention of α-synucleinopathies awaits further investigation, in particular multicenter double-blind trials.

## 1. Introduction

Rapid-eye-movement (REM) sleep behavior disorder (RBD) is a REM sleep parasomnia that shows abnormal motor behaviors emerging during REM sleep in a context of diminished normal hypotonia [1]. Various visual disturbances are associated with synucleinopathies [2]. It has been established that isolated RBD (iRBD) is a reliable prodromal manifestation of any α-synucleinopathy within 20 years of onset of iRBD [3]. Melatonin, a chronobiotic methoxyindole with a well-known cytoprotective and antioxidant and free radical scavenging properties, has shown effectiveness for symptomatic treatment of RBD [4]. This is a necessary intervention to avoid the risk of injury to oneself or one’s bed partner and also to avoid the sleep disruption that these episodes entail. In addition, in the absence of specific treatments, patients with iRBD are candidates for neuroprotection trials or even better, disease-modifying treatments, that delay or prevent conversion to a pathology with abnormal α-synuclein metabolism [5]. This review will discuss two different aspects related to the action of melatonin in RBD: 1. As a symptomatic treatment in RBD and 2. As a possible disease-modifying treatment in α-synucleinopathies.

## 2. Circadian Modulation of REM Sleep

Sleep is regulated by two interacting processes: circadian and homeostatic processes [6]. The circadian process is driven by the body’s internal biological clock. The circadian process promotes wakefulness during the day and sleepiness at night, with a peak in sleepiness occurring during the late night and early morning hours. The second process is the homeostatic process, which is driven by the body’s need for sleep and is based on the concept of sleep debt. The homeostatic process monitors the quality and amount of an individual’s sleep and adjusts the drive for sleep accordingly. The longer an individual is awake, the greater the homeostatic sleep pressure, leading to increased sleepiness and a stronger drive to sleep [6]. Timing and duration of sleep interact. The circadian process regulates the timing of sleep and wakefulness, while the homeostatic process regulates the duration and depth of sleep. Together, these processes help to maintain a balance between sleep and wakefulness, promoting healthy and restorative sleep [6]. Neurodegenerative processes include early disruption of sleep and circadian rhythms, and both types of disturbances are recognized as key factors in neurodegeneration [7,8]. RBD gives a window for insight into the early development of α-synucleinopathies, i.e., Parkinson’s disease (PD), multiple system atrophy and Lewy body dementia.

Circadian oscillations are based on interconnected transcriptional and post-translational feedback loops that are regulated by core clock genes [9]. The transcriptional/translational feedback loops comprising the core clockwork were studied by transgenic gene deletion technology in rats and mice. Delays in the feedback loops, partly depending on phosphorylation of the clock proteins that affects their stability and transcription complex formation, are the basis of the 24 h oscillation in clock gene expression [10]. As the Clock and Bmal1 genes in mammals are transcribed, the transcription factors BMAL1 and CLOCK are produced, which then dimerize via basic helix-loop-helix (bHLH) domains. The dimers subsequently stimulate the transcription of Cry and Per, the proteins CRY and PER inhibiting the expression of BMAL1 and CLOCK. The cycle is restarted as PER and CRY decline over time [10]. Per and Cry mRNA levels in the suprachiasmatic nucleus (SCN) have a maximum in middle/late afternoon [11]. Bmal1 mRNA levels rise around midnight, but Clock mRNA is present in the SCN at a constant concentration all day [12]. Production of PER and CRY is limited by binding to the E-box element of the promoter regions of Bmal1, Clock, Rev-Erb and other clock-controlled genes via the CLOCK/BMAL1 complex [10]. For translocation to the nucleus, PER and CRY are phosphorylated by casein kinase 1 δ/ε [12]. Four secondary regulatory loops modulate master oscillation further. A first one involves ROR (retinoid-related orphan receptor) and the nuclear receptors REV-ERB. REV-ERB suppresses Bmal1, whereas ROR promotes it by attaching to the RORE (response element-binding site) sequence in the Bmal1’s promoter region [13,14]. A second regulatory loop is given by the protein ROR that also binds to the RORE element in the promotor of CLOCK, and to NFIL3 (nuclear factor, interleukin 3 regulated), thus inducing their transcription. NR1D1 (nuclear receptor subfamily 1 group D member 1) and possibly other proteins from this family inhibit ROR binding to the RORE element. A third regulatory loop comprises DBP (D-box binding PAR b ZIP transcription factor), a protein whose expression is controlled by BMAL1:CLOCK from the first loop, which binds to the D-box in the promotor region of PER. NFIL3 from the second loop regulates this binding negatively [10]. A fourth regulatory loop is given by DEC (AKA Basic helix-loop-helix family member e40 loop) as an ancillary circadian loop characterized by the expression of DEC and other circadian-controlled genes under regulation by BMAL1:CLOCK. DEC, in turn, inhibits BMAL1:CLOCK binding to the E-box element, regulating its expression [15].

In normal REM sleep, two neuronal networks, one suppressing motor-skeletal activity and the other generating muscle atonia, are involved [16]. Active inhibition by neurons in the medulla explains the muscle atonia. The thalamus influences spinal motor neurons, and locomotion implicates input from the forebrain. The laterodorsal tegmental nucleus, the pedunculo-pontine nucleus and the peri-locus coeruleus region are brainstem regions involved in RBD pathophysiology [17]. REM atonia is handled by supra-spinal mechanisms. Muscle atonia in REM sleep depends on the stimulation by nuclei in the pons and medulla, which then transmit descending inhibitory projections to hyperpolarize spinal α- motoneurons. Obliteration of this mechanism leads to muscle activity during the REM sleep [18]. At the early stages of α-synucleinopathies, other sleep characteristics besides REM associated atonia remain relatively unaffected.

Under usual entrained conditions, the probability of REM sleep to occur (i.e., REM sleep propensity) is highest during the second half of night sleep [19]. A circadian profile characterizes REM sleep, which is more dependent on the circadian phase than on previous wakefulness duration [20]. Almost every aspect of REM sleep is modulated by the circadian system. However, these parameters are not altered in iRBD. Animal research supports such a circadian modulation in SCN lesioned rats, in which this circadian modulation of REM sleep vanished [21]. Experiments with crepuscular mammals indicate that both diurnal and nocturnal REM sleep deprivation lead to comparable REM sleep debt, but only after nocturnal deprivation does a consistent REM sleep rebound occur, indicating active promotion of REM sleep by the circadian system [22].

The therapeutic effect of melatonin is another indication on the involvement of the circadian system in α-synucleinopathies. Kunz et al. have proposed that a reason for controversial results lies on the specific mode of action of melatonin as a chronobiotic [23,24]. Melatonin is commonly used as a conventional sleeping aid, administered 1–2 h prior to bedtime. However, in several studies, administration of melatonin in a chronobiotic protocol at the moment of expect maximum effect (around 0100 h) adjusted for chronotype, exerts a gradual therapeutic effect on RBD seen within 1 to 2 weeks [23,24]. α-Synucleinopathies are seen as a complex combination of motor and non-motor disturbance preceded by a prodromal, non-motor phase including alterations of sleep, autonomic disorders, sensory deficits and cognitive impairment, among others [8,25]. Since diurnal fluctuations are detected in non-motor symptoms during the course of the disease, it is logical to associate the symptom fluctuations with chronodisruption [26].

An example is given by PD [27]. Both during the course of the disease, as well as after dopaminergic treatments used to mitigate Parkinsonian symptoms, clock genes and melatonin significantly change [28,29]. In addition to be the best prognostic biomarker for the development of α-synucleinopathies, iRBD represents the ongoing neurodegenerative process itself [3,30]. The rhythmicity of circadian clock genes and circulating melatonin levels were measured in iRBD patients [31]. Abolition of circadian rhythmicity of Bmal1, Per2 and Nr1d1, intactness of rhythmicity of Per1, and a reduced amplitude of Per3 rhythm were documented in iRBD patients. No change in the diurnal melatonin rhythm was observed. RBD patients had a more dispersed range of acrophases of melatonin rhythm (>11 h) as compared to the control group exhibiting stable acrophases with approximately 5 h dispersion. A phase delay of about 1 h was seen in iRBD patients as compared to controls [31].

Disturbances of biological rhythms emerge in fully developed α-synucleinopathy. For example, PD patients exhibited lower expression of BMAL1 [32,33,34]. In a sample of 1253 Chinese PD patients vs. 1342 controls, polymorphism in PER1 and ARNTL genes were uncovered reinforcing the link of circadian alterations with α-synucleinopathies. Symptomatic severity of RBD correlated with an altered body temperature rhythm [35]. Disturbances in blood pressure or cortisol circadian rhythms were also observed [36,37]. In the sleep study of the longitudinal cohort Osteoporotic Fractures in Men Study (comprising more than 2900 men), a higher risk of incident α-synucleinopathy was associated with disrupted circadian rhythmicity [38]. Non-REM sleep symptoms remain poorly understood in α-synucleinopathies [39].

## 3. RBD as a Prodrome of α-Synucleinopathies

As mentioned, α-synucleinopathies are associated with RBD or daytime sonmnolence years in advance to a diagnosis based on motor, behavioral or autonomic disturbances. Sleep deprivation leads to impairment of clearance of waste proteins by the glymphatic system with concomitant neurodegeneration. In a recent study, glymphatic system activity, evaluated by diffusion tensor image analysis along the perivascular space (ALPS), indicated a lower ALPS index (impairment of glymphatic system activity) in PD and iRBD patients as compared to healthy controls. The ALPS index and elevated disease severity were negatively correlated in the iRBD and PD subgroups [40]. The recently described subarachnoid lymphatic-like membrane in murine and human brains provides fundamental insights into brain fluid transport and immune barriers and opens questions about its function in the presentation of neurodegenerative diseases and related sleep disorders [41].

iRBD has proven to be a robust predictor of disease since more than 80% of iRBD patients will develop some α-synucleinopathy with a conversion rate of 6.3% of diagnosed patients by year [41,42]. On clinical grounds, the first observation corresponded to James Parkinson that pointed out the presence of excessive motor activity during sleep resembling subsequent descriptions of RBD in 1817 [43]. He wrote: “In this stage the sleep becomes much disturbed. The tremulous motion of the limbs occurs during sleep, and augments until they awaken the patient, and frequently with much agitation and alarm”. In modern science, the first observation of loss of atonia during REM sleep was that of Jouvet and co-workers [44,45] followed later on by the contributions of Hendricks et al. [17]. The complete description of the phenomenon in humans was made by Schenk and colleagues [46]. 

Dream-enactment motor behaviors that emerge during episodes of REM sleep define RBD as a parasomnia. Two types of RBD occur, i.e., isolated or symptomatic. In the latter case it can be a comorbidity related to neurological disorders like neurodegenerative diseases, or narcolepsy, to autoimmune or paraneoplastic diseases, or to substance abuse or withdrawal. It is important to highlight that not all of the patients with α-synucleinopathies have RBD. It is found in 50% of PD, 90% of multiple system atrophy and 80% of dementia with Lewy bodies. It is recommended that once patients are identified with isolated RBD it is mandatory to stratified them in conjunction with other markers of disease since the diagnosis of iRBD is not sufficient to distinguish subtypes of α-synucleinopathy [47]. Despite these limitations, iRBD constitutes a robust prodromal marker of α-synucleinopathies and to date, the best biomarker available to predicts α-synucleinopathy diseases [48], with the following “pros”: (1) A prolonged prodromal time, (2) An excellent predictive power and when diagnosed as iRBD, and (3) The absence of disease-related treatments that could act as confounders [42,49].

## 4. Basic Physiology and Biochemistry of Melatonin Relevant to RBD

Two major functions are ascribed to melatonin and are relevant for the neurodegeneration. Melatonin is a chronobiotic and at a higher dose, an effective cytoprotector. Thus to have a full view of melatonin utility in α-synucleinopathies both aspects should be considered. Melatonin is the prototype of the endogenous signals regulating the circadian system (the so-called “chronobiotics”) [50,51]. The sleep/wake cycle in both normal and blind patients is strongly related to the circadian rhythm of melatonin in blood [52]. The association of plasma melatonin levels with the circadian processes that govern sleep proclivity is clearly established [53,54]. A fundamental information on the time of year regulating neuroendocrine seasonality depends on the secretion of pineal melatonin which is proportional to night duration [55].

Circulating melatonin is produced almost exclusively by the pineal gland. As soon as it is synthesized, melatonin diffuses out into the capillary blood [56] and cerebrospinal fluid (CSF) [57]. Melatonin is found earlier in the third ventricle than in the lateral ventricle CSF. Levels of melatonin in ventricle CSF were up to 30 times higher than circulating levels [57], whereas spinal CSF values were in the range of those found in blood [58]. Hypothalamic melatonin concentrations up to 50 times greater than in plasma were described by high pressure liquid chromatography [59] or radioimmunoassay [60]. This indicates the existence of two compartments of melatonin affecting physiological function, i.e., in CSF affecting neurally mediated functions and in plasma acting on peripheral organs. Circulating melatonin derived from the pineal gland is about 5% of the total melatonin produced in the body. Peak concentration of circulating melatonin occurs at night with maximum at a younger age [61]. Increased levels of oxidative stress and associated degenerative changes seen in α-synucleinopathies are probably related to the age-associated decline in melatonin production [62]. Melatonin is synthesized locally in most cells [63] and the hypothesis that it is produced in all animal cells that have mitochondria [64] and that this mitochondrial function is critical for cytoprotection [65] is widely accepted.

The chronobiotic function of melatonin relies mainly on melatonergic receptors (MT1 and MT2), which belong to the superfamily of membrane receptors associated with G proteins [66]. GPR50 is another member of the melatonin receptor family which rather than binding melatonin it forms homo and heteromers with MT1 and MT2 [67]. The SCN and several other CNS areas, like the cerebral and cerebellar cortex and the midbrain, display MT1 and MT2 receptor activity [68]. Substantia nigra, caudate-putamen, ventral tegmental areas and nucleus accumbens contain melatonin receptors [69]. In the case of substantia nigra MT1 and MT2 receptors, a depressed signal was found in PD patients [70]. Cell membranes are crossed easily by melatonin due to its amphiphilic properties. In the cytoplasm melatonin interacts with proteins like calmodulin and tubulin [71]. In the nucleus, indirect interaction of melatonin with orphan RZR/ROR superfamily of receptors [65] via the activation of sirtuin-1 has been detected [72]. 

However, interaction with MT1 and MT2 receptors cannot explain fully the cytoprotective effect of the methoxyindole. The melatonin doses needed to modify the intracellular melatonin are significantly higher than those used clinically as a chronobiotic [73]. Although in cell cultures, nanomolar concentrations of melatonin are effective [74], pharmacological doses largely exceeding receptor saturation are needed for cytoprotection [75]. Excellent summaries on the activity of melatonin to reverse altered signaling mechanisms in neurodegeneration have been published [76,77,78]. The antioxidant effects of melatonin on free radical production are largely independent of receptors. In addition to be a free radical scavenger itself, a cascade of compounds with higher antioxidant activity is produced after melatonin oxidation. As an indirect antioxidant, melatonin also promotes the synthesis of antioxidant enzymes while inhibiting that of pro-oxidant enzymes [79]. Under ischemic conditions (not related to free radicals) the stabilizing activity of melatonin on mitochondrial membranes explain its antiapoptotic and cytoprotective effects [64]. 

The mitochondrial aspects of α-synucleinopathies that arise from the activity of α-synuclein as an excitotoxin must be considered when discussing melatonin activity [80,81]. They include free radical generation and calcium overload-mediated changes in mitochondrial membrane potential and in permeability transition pore (mtPTP). Mitochondria are not only a major site of radical oxygen species (ROS) generation, but also the primary target of attack for ROS and radical nitrogen species (RNS) [82]. A vicious cycle leading to further generation of free radicals arises after damage of the mitochondrial respiratory chain via breakdown of the proton potential and opening of the mtPTP; apoptosis is thus elicited. Indeed, mitochondrial effects explain the neuroprotective role of melatonin in α-synucleinopathies. Besides ROS and RNS scavenging, additional actions of melatonin are stimulation of glutathione (GSH) synthesis, support of the GSH disulfide (GSSG) reduction and protection of mitochondrial membranes and DNA from oxidative insults [83,84]. Some others are indirect antioxidant effects of melatonin, like the maintenance of mitochondrial electron flux [85,86]. In contrast to other antioxidants, the balanced amphiphilicity of melatonin allows crossing of the cell membranes to be concentrated within mitochondrial compartments [86,87,88]. Melatonin administration increased the activities of mitochondrial respiratory complexes I and IV in the brain [89]. Melatonin antagonizes mitochondrial protein misfolding damage given by free radicals and changes in membrane potential due to overexcitation caused by excess calcium.

The brain has a high oxygen consumption rate and is relatively rich in polyunsaturated fatty acids which can be peroxidized under oxidative stress [90]. Lipid peroxidation leading to oxidative protein modifications can be effectively suppressed by melatonin, particularly in the central nervous system [91]. Melatonin is very effective to scavenge free-radicals like the extremely reactive hydroxyl radical, carbonate radicals and RNS [92]. Metabolites of melatonin, like cyclic 3-hydroxymelatonin, N1-acetyl-N2-formyl-5-methoxykynuramine (AFMK) and N1-acetyl-5-methoxykynuramine (AMK) also share the activity of the mother molecule [84,93]. GSH peroxidase was consistently upregulated by melatonin in the brain [84,93]. Down-regulation of prooxidant enzymes such as lipoxygenases and nitric oxide (NO) synthases is also seen after melatonin. Thus, by avoiding peroxynitrite-derived radicals and NO-dependent neuronal excitation, and by inhibiting inflammatory reactions, the oxidative and nitrosative damage is attenuated by melatonin [94]. Moreover, melatonin decreases Pp53, Bax, and caspase 9 expression [95] and increases Bcl-2 and p53 levels [96] leading to the inhibition of the apoptosis pathway. Microglial activation in neuroinflammation plays a causative role in α-synucleinopathies exacerbating the pathological consequences of the disease. Increased inflammatory factors including nuclear factor κB (NF-κB), IL-1, IL-6, cyclooxygenase (Cox)-2, TNF-α, iNOS, and INF-γ in glial cells, and elevated oxidative stress due to excessive free radical generation following mitochondrial damage, play essential roles in the progression of α-synucleinopathies [97,98].

Dopaminergic neuron loss and α-synuclein accumulation is found in 1-tmethyl-4-phenyl-1,2,3,6-tetrahydropyridine (MPTP)-induced murine model of PD. Death of dopaminergic neurons depends on toll-like receptor 4 (TLR4) stimulation via cytokine release evoked by NF-κB in activated glial cells. In the MPTP mouse model melatonin reduces dopaminergic neuron loss, decreases cytokine release due to the inflammatory response and suppresses TLR4-mediated neuroinflammation [99]. Melatonin displays both pro-inflammatory and anti-inflammatory effects [100,101]. The inhibition of COX [102] (mainly Cox-2) [103] and of NF κB binding to DNA, and the down-regulation of inducible NO synthase receptors, underlie melatonin anti-inflammatory activity [65]. Other anti-inflammatory pathways include prevention of NLRP3 inflammasome activation and up-regulation of nuclear factor erythroid 2-related factor 2 [65]. Ultimately, these effects lead to an augmented production of anti-inflammatory cytokines and decreased levels of pro-inflammatory cytokines [100,101].

Brain magnetic resonance spectroscopy in PD patients documented a γ-aminobutyric acid (GABA)-ergic dysregulation in basal ganglia [104]. Dopaminergic neurodegeneration probably led to a GABA hypofunction in basal ganglia because striatal dopaminergic axons release GABA [105]. Administration of the GABA agonist baclofen in a murine model of PD ameliorated motor symptoms and protected dopamine cell bodies [106]. Besides its anti-excitatory and sedative effects [107,108] melatonin may act on the GABAergic system to mediate neuroprotection. Through the activation of GABAergic receptors melatonin protects neurons from ß-amyloid peptide toxicity [109]. The efficacy of the benzodiazepine antagonist flumazenil to modify upregulation of GABA activity by melatonin, and the lack of activity of the melatonin receptor antagonist luzindole, support the occurrence of an allosteric modulation of GABAA receptors by melatonin [110]. Melatonin has a potent anti-excitotoxic activity. Melatonin administration prevents CA1 neuron death in the hippocampus after transient ischemia [111] or due to high doses of glucocorticoids [112]. The neuronal death induced by the agonist of the ionotropic glutamate receptor kainite is also prevented by melatonin [113]. Neither MT1 nor MT2 melatonin receptor blockers are able to modify this anti-excitotoxic activity of melatonin [114].

Of importance in relation to α-synucleinopathies is the activity of melatonin to counteract an increased insulin resistance (IR) [115]. Both brain and systemic IR are documented in PD [116,117] and type-2 diabetes (T2D) is a risk factor aggravating PD development [118]. The pathological signs of PD are Lewy bodies and neurites, composed of amyloid aggregates of misfolded α-synuclein [119,120]. Aggregation of islet amyloid polypeptide (amylin) is found in the pancreas in T2D, the deposit of amylin accelerating the formation of α-synuclein amyloid [119]. direct interaction between amylin and α-synuclein was indicated by studies in pancreatic ß-cells of patients with a neuropathological diagnosis of α-synucleinopathy [121]. In experimental models of PD, melatonin increases the concentration of nigral and striatal dopamine [122]; it also reportedly prevents the depletion of dopamine and disruption of dopaminergic neurons [95] and neurotoxins-induced dopaminergic neuron death [123]. An elevation in the activity of superoxide dismutase, catalase, and GSH peroxidase and reduction in the malondialdehyde level and death of dopaminergic neurons were documented after melatonin treatment in the substantia nigra of rats with PD model induced by 6-hydroxydopamine [124]. Administration of melatonin also reduces oxidative stress in the MPTP murine model of PD [125]. In view of melatonin’s capacity to suppress free radicals, transfer electrons, and repair damaged biomolecules, it may effectively protect neurons and glial cells from the oxidative stress pathway in α-synucleinopathies. In MPTP-induced PD mice, melatonin prevents the rise of iNOS, as a pathologic hallmark of neuroinflammation [126]. The protective effect of melatonin on α-synuclein-induced damage to dopaminergic neurons in the substantia nigra has been observed in animal models [123]. Melatonin prevents α-synuclein assembly and fibril formation by suppressing protofibril development and instability in precursor fibrils. 

The expression of aquaporine-4 (AQ4) is significantly reduced in PD patient brains compared to the healthy individuals [127] The AQ4 water channels have an essential role in lowering CSF α-synuclein levels [128]. Melatonin preserves the function of glymphatic system and increases AQ4 expression [129,130]: hence, it has a favorable effect on PD patients. However, further clinical trials are required to definitively prove the beneficial effects of melatonin on PD patients [131] (Figure 1).

## 5. Melatonin and Melatonin Analogs in RBD

Table 1 summarizes data on the use of melatonin and melatonin analogs to treat RBD. In 1997 the first observation on melatonin as a treatment was published [133]. A 64-year-old man with clinically and polysomnographically confirmed RBD was examined. He was also an insomniac with excessive daily somnolence and short time memory problem and did not receive clonazepam because of comorbidities. After melatonin administration (3 mg p.o./daily at bedtime) the RBD symptoms, cognitive/mnestic deficits and sleep abnormalities showed a complete clinical recovery [133].

The findings were confirmed in the same laboratory in an open-labeled trial with 6 patients (melatonin 3 mg daily) [24] and also in a randomized, double-blind, placebo-controlled trial in a crossover design with 8 patients (placebo or melatonin 3 mg daily, each for a period of 4 weeks [23]. Video-polysomnographic examination (vPSG) was used in all cases to confirm RBD. Twelve out of the 14 patients improved clinically and showed significantly decreased REM sleep without atonia (RWA) in vPSG (Table 1).

In a prospective observational study, the effect of melatonin (3–9 mg daily) was examined in 15 patients with vPSG-confirmed RBD [134]. A mild to strong improvement of symptoms was reported by 13 patients. Melatonin administration significantly reduced percentage of tonic REM activity in vPSG. Baseline melatonin levels were low in the presence of a higher response in RBD patients. 

The efficacy of melatonin as monotherapy and in combination with clonazepam was examined in a retrospective observational study [135]. Fourteen RBD patients were treated with melatonin (3–12 mg daily). Coexisting comorbidities were PD, narcolepsy, dementia with Lewy bodies and multiple system atrophy. RBD was controlled by melatonin in 6 patients, significantly improved in 4, and transiently improved in 2. Continued benefit with melatonin beyond 12 months of therapy was seen in 8 subjects [135]. 

Anderson and Shneerson [136] reviewed 39 patients with confirmed RBD. In 58% of the patients using clonazepam adverse effects were reported. Twenty-one patients continued to take clonazepam, 8 used another medication, and 4 required a combination of medications to control symptoms adequately. Two patients successfully used 10 mg melatonin p.o./daily. A combination therapy (lonazepam/gabapentin/melatonin) was used and found effective [136].

A survey of 45 consecutive RBD patients seen at Mayo Clinic between 2008–2010 was published [137]. The primary outcome variables were RBD visual analog scale (VAS) score and injury frequency. Twenty-five patients received melatonin (6 mg/daily), 18 received clonazepam (0.5 mg daily) and 2 received both as initial treatment. RBD VAS ratings were significantly improved after treatments. Significantly reduced injuries and fewer adverse effects were observed in melatonin-treated patients [137]. 

Another retrospective analysis evaluated the data from 28 patients with PSG-confirmed RBD [138]. Comorbidities observed were PD (10 patients) and cognitive decline (4 patients). All patients received melatonin (3–6 mg daily). After 4 months treatment with 6 mg melatonin nightly, 26 patients showed a clear clinical improvement. Following the first period of melatonin monotherapy, all patients received a combination therapy with clonazepam 0.5–1 mg per night. A significant reduction in the percent of wakefulness after sleep onset was found in the patients with 6 mg melatonin first and subsequently combined with clonazepam [138,139].

Obstructive sleep apnea (OSA) is known to provoke RBD-like symptoms and both sleep-related pathologies usually coincide because of overlapping prevalence in similar age groups. In a case series study on four patients with a severe clinical RBD syndrome and concomitant OSA, daily administration of 2 mg prolonged release melatonin resulted in improvement of RBD symptoms but persistence of the sleep related breathing disorder [140]. REM sleep without atonia values in PSG with melatonin were abnormally high, probably because of the untreated OSA. indicating the necessity of treatment of both disorders for an optimal therapeutic response.

In a case report study, a 72-year-old man was clinically diagnosed as RBD in 2011 [141]. A reduced DA transporter density typically indicating an impending PD was diagnosed by DA transporter scintigraphy (DaTSCAN). After 6 months of daily melatonin treatment (2 mg prolonged release) the clinical signs of RBD disappeared. A control PSG in 2014 confirmed the presence of a normal REM sleep. Additional DaTSCANs were performed in 2013 and 2015. Compared to the clear PD signs in the 2011 scan, the 2013 scan was borderline and the 2015 scan was without any sign of PD. The authors interpreted the results as a possible neuroprotective role for melatonin in α-synucleinopathy [141].

A single-center, observational cohort study of 209 consecutive iRBD patients treated with melatonin was undertaken to assess treatment effects, time course and confounding factors [4]. A total of 171 patients received 2 mg prolonged release melatonin (2 mg, ≥6 months), 13 were under melatonin for about 1–3 months, and 25 received mixed treatments. RBD symptomatology gradually improved over the first 4 weeks of treatment with melatonin and remained stably improved for up to 21.7 years [4].

It must be noted that negative results with melatonin (prolonged release) have also been reported. In a randomized, double–blind, placebo–controlled pilot study on 30 iRBD to assess the effects of prolonged-release melatonin (2 or 6 mg/day), primary outcomes (scores from the Clinical Global Impression-Improvement and the Korean version of the RBD questionnaire-Hong Kong) and secondary outcomes (Pittsburgh Sleep Quality Index score, the Epworth Sleepiness Scale score, and the frequency of dream–enacting behaviors) remained unchanged after melatonin administration [142].

A dose of 4 mg of prolonged-release melatonin or placebo p.o. once-daily before bedtime was given to 30 PD patients with RBD in a double-blind, placebo-controlled trial [143]. No differences between melatonin-treated and placebo were found. The authors concluded that prolonged-release melatonin 4 mg did not reduce RBD disorder in PD [143]. Presumably, melatonin at low dose is relatively ineffective when RBD is co-morbid with PD or OSA [140]. 

In a prospective, open-label, randomized trial undertaken to compare efficacy and safety of melatonin and clonazepam, RBD patients received either clonazepam 0.5 mg or prolonged-release melatonin 2 mg at bedtime for 4 weeks [144]. In 34 patients with probable RBD, scoring parameters of RWA improved after clonazepam treatment but not after melatonin treatment. Daytime sleepiness and insomnia symptoms were reduced by melatonin but not by clonazepam. The proportion of N2 sleep was increased, and N3 and REM sleep were decreased only in the clonazepam group. Depressive symptoms increased after clonazepam [144]. 

Concerning melatonergic agonists only a limited number of studies examined their efficacy in RBD. Three publications examined the effect of ramelteon. In an open-labeled trial, 12 consecutive patients with idiopathic RBD were treated for at least 4 weeks with 8 mg ramelteon 30 min before bedtime [145]. Ramelteon treatment did not have any effect on RBD severity scale or REM sleep without atonia measured by vPSG.

Two patients with secondary RBD complications along with neurodegenerative diseases including multiple system atrophy and PD received ramelteon (8 mg per night) in monotherapy due to contraindications to clonazepam [147]. Ramelteon treatment improved RBD severity scale or REM sleep without atonia measured by vPSG.

In another study 35 patients from multiple centers with idiopathic PD comorbid with sleep disorders were evaluated for ramelteon response [146]. The patients received 8 mg of ramelteon before sleep once daily for 12 weeks. Twenty-four out of the 35 patients examined were diagnosed with probable RBD (PRBD) using the Japanese version of the RBD screening questionnaire. Sleep disruption in PD patients was curtailed by ramelteon [146]. 

Concerning agomelatine, a case review described its positive effects in 3 patients with clinical and PSG confirmed iRBD [148]. Aggressive behavior fully remitted in 1 patient after treatment with 25 mg per night. The other 2 patients received 50 mg per day agomelatine with clearly reduced numbers of RBD episodes. 

A systematic review of randomized controlled trials and observational studies that addressed interventions for the management of RBD was published on behalf of The American Academy of Sleep Medicine [149]. The study concluded that the overall certainty of evidence for melatonin to treat iRBD was low. Clinical improvement in decreasing frequency and/or intensity of dream enactment episodes were noted among patients taking immediate-release melatonin and to a lesser degree prolonged-release melatonin. In a critical review Gilat et al. concluded that the effectiveness of the two first-line therapies for RBD (melatonin and clonazepam) are probably overestimated [150].

The differences in efficacy of fast release and prolonged release melatonin may depend on the concentrations attained at the phase delay and phase advance portions of the night. Although it is widely accepted that natural melatonin’s chronobiotic influence is mediated by MT receptors, a chronobiotic effect can also be observed when pharmaceutical amounts of fast-release melatonin (that saturate receptors) are utilized. Even at such high doses, melatonin ingested as a fast-release preparation at a single time of day (bedtime) maintains chronobiotic effects contrasting with the more or less similar pharmacological concentrations during the whole night attained by prolonged release melatonin [151].

## 6. Concluding Remarks

Utilizing preclinical models has significant potential to advance the understanding of the interplay of α-synucleinopathies with the circadian system and sleep. In melatonin research, animal models has been widely studied but many of their findings are overlooked. In particular the translation doses calculated by allometry [152] is generally disregarded to discuss melatonin use in humans, regardless than these studies are very useful to calculate the initial doses of compounds use in clinical Phase 1 studies. The effect of melatonin in animal models of α-synucleinopathies indicates that, from the melatonin doses used in each case, a human equivalent dose (HED) of melatonin is in the 100 mg/day range [132]. 

Dosage escalation trials showed melatonin’s absence of toxicity in humans in doses up to 100 mg [153,154]. Melatonin has a high safety profile and is generally well tolerated (see ref. [75]). In the USA, the number of individuals aged >65 years who have used melatonin in the last month increases 3-fold over the last two decades [155]. Typically few, mild to moderate in intensity, and either self-limiting or resolved promptly after treatment discontinuation, adverse effects of melatonin have been documented [156,157,158].

In the United States an estimated 3.1 million individuals (1.3 percent of adults) consume melatonin daily [159]. The manufacturing quality and bioavailability of melatonin and the potential contaminants are questionable in these unlicensed melatonin preparations [160,161]. In that aspect, commercial melatonin labeled with the U.S. Pharmacopeia (USP) Verification Mark may provide the most consistent dosing among treatment options.

To what degree melatonin has therapeutic efficacy in the prevention or treatment of α-synucleinopathies awaits further investigation, in particular multicenter double-blind trials. Because of the HED of melatonin determined from preclinical studies, melatonin dosages need to be reviewed. Indeed, given the number of scientific/medical papers that have recommended its use, melatonin’s failure to garner attention as a potential treatment for α-synucleinopathies is disappointing. The fact that no significant group has advocated for its therapeutic usage in treating this condition is one of several potential causes for this. 

The pharmaceutical business is not motivated to promote the use of melatonin because it is not patented and affordable. Nonetheless, it would be wise for the pharmaceutical business to research the possibility of a profitable and medically effective combination of melatonin with other medications. Due to its low cost, minimal toxicity, and ability to be taken orally, melatonin would be particularly advantageous. This is particularly true in underdeveloped nations where individuals have less money to spend on treating age-related α-synucleinopathies.

## Figures and Tables

**Figure 1 brainsci-13-00797-f001:**
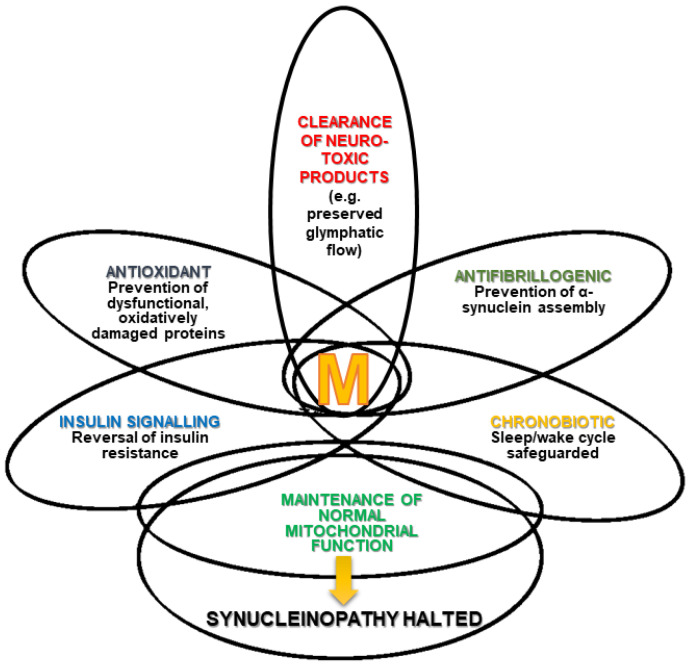
Melatonin (M) and synucleinopathies. The Figure depicts the multiple effects of melatonin and the different degree of overlap (interrelations and mutual influences) they have (modified from [132]).

**Table 1 brainsci-13-00797-t001:** Studies including RBD patients treated with melatonin or melatonin receptor agonists.

Subjects and Concomitant Disease	Design and Diagnostic Criteria	Study’s Duration	Treatment	Measured	Number of Patients Responding	PSG	Ref.
64 year-old male iRBD patient	Case report	5 months	3 mg melatonin p.o./daily at bedtime	Actigraphy, PSG	Actigraphy indicated a decrease in motor activity during sleep.	↑ % REM-Sleep, ↓ RWA, ↓ phasic electromyographic activity in REM	[133]
6 RBD patients, 50% males, mean age 54 years, 1 PD, 3 with memory and concentration deficits	Open-label prospective case series	6 weeks	3 mg melatonin p.o./daily at bedtime	PSG	In 5 patients, improvement of symptoms was observed during treatment and extended for weeks or months after treatment.	↑ REM atonia, ↓ stage shifts in REM epochs	[24]
8 male RBD patients, mean age 54 years, 2 narcolepsia, 1 PD	Double blind, placebo-controlled trial	4 weeks	3 mg melatonin p.o./daily at bedtime.	PSG	Improved clinical global impression in 7 patients.	↓ frequency of RBD episodes ↓ 30-s epochs of RWA	[23]
14 male RBD patients, mean age 63.5 years, RBD with no further disclosures	Open-label prospective case series	variable	3–9 mg melatonin p.o./daily at bedtime	PSG	Normalized problem sleep behavior in 13 patients after melatonin treatment. Patients with low melatonin levels tended to respond to melatonin therapy better.	↓ % tonic REM activity after melatonin administration	[134]
13 male RBD, onset age 56 years, 7 DLB, 2 mild cognitive impairment, 2 narcolepsia, 1 PD	Retrospective case series	14 months	3–12 mg melatonin p.o./daily at bedtime, +clonazepam 0.5–1 mg in 7 patients	Clinical Global Impression	After 12 months of treatment, 8 patients had improved symptoms of RBD.	No	[135]
38 Male RBD patients, mean age 66 years 1 mild cognitive impairment + mild OSA	Retrospective case series		All initially treated with clonazepam. When melatonin was used, it was given at a 10 mg p.o./daily at bedtime.	Clinical Global Impression	To control symptoms, 21 patients received clonazepam, 8 used another medication, and 4 needed a combination of adequately. Melatonin was successfully used in 2 patients. Combination therapy (lonazepam/gabapentin/melatonin) was used in 1 patient.	No	[136]
45 RBD patients. 77.8% Male, mean age 66 years	Retrospective case series	27–53 months	In 25 patients, 6 mg melatonin p.o./daily at bedtime. In 18 patients 0.5 mg clonazepam monotherapy	Clinical Global Impression	89% improvement on clonazepam, 68% improvement on melatonin. Melatonin ↓ injuries significantly. Patients receiving melatonin reported fewer adverse effects.	No	[137]
28 patients, 72% male, 66.5 years mean age, 10 PD, 16 with OSA, 6 with cognitive decline	Retrospective review	4 months	melatonin 3–6 mg per night for 4 months, +0.5–3 mg clonazepam	PSG	26 with 6 mg melatonin in monotherapy.	↓ % tonic REM activity and wake after sleep onset	[138]
203 consecutive patients with iRBD	Retrospective case series	4 years	6 mg melatonin as a median dose (range 1.9–9 mg), 1 mg clonazepam as a median dose (range, 0.25–4 mg)	PSG	Melatonin treatment effective in 32 patients (15.7%). Clonazepam was replaced by melatonin In 24 subjects who experienced side effects or in whom clonazepam was not effective, melatonin was given instead.	↓ % tonic REM activity after melatonin or clonazepam administration.	[139]
4 RBD patients with concomitant obstructive sleep apnea	Open label	4 weeks	2 mg prolonged release melatonin p.o./daily at bedtime	PSG	Treatment led to a relevant clinical improvement of RBD symptoms in all patients.	REM without atonia incidence was high probably because of the untreated comorbid condition	[140]
72 year-old male iRBD patient	Case report	5 years	2 mg prolonged release melatonin p.o./daily at bedtime	PSG and DA transporter scintigraphy (DaTSCAN)	In 2011, the patient was clinically suspected of PD. DaTSCAN revealed reduced DA transporter density and PSG confirmed the diagnosis of RBD. After 6 months of melatonin treatment, clinical signs of RBD were absent.	Control PSG in 2014 indicated normalized REM sleep with atonia. Additional DaTSCANs were performed in 2013 and 2015 indicated normalization of DA transporter density	[141]
30 iRBD patients, 66% male, mean age of 66.4 years	Double blind, placebo-controlled trial	4 weeks	2 mg or 6 mg prolonged release melatonin o placebo p.o./daily at bedtime	Korean version of the RBD questionnaire. Impression Improvement (Clinical Global Impression)	Neither primary nor secondary outcomes examined were affected by treatment.	No	[142]
209 RBD patients	Single-center, observational cohort study	171 patients ≥6 months, 13 patients 1–3 months, 25 patients received a mixed treatment	171 patients took melatonin (2 mg prolonged release), ≥6 months, always-at-the-same-clock time, 10–11pm, corrected for chronotype), 13 patients had applied melatonin for about 1–3 months, and 25 patients underwent mixed treatments	1529 clinical evaluations performed, including Clinical Global Impression and RBDsymptom severity scale (Ikelos-RS), analyzed using linear mixed models	With melatonin, RBD symptomatology regressed gradually up to 4 weeks of treatment and remained improved thereafter. When melatonin was discontinued after 6 months, symptoms remained stably improved. When administered for only 1–3 months, RBD symptoms gradually returned.	Video-PSG-confirmed RBD	[4]
30 RBD patients	Randomized, double-blind, placebo controlled, parallel-group trial	8 weeks	4 mg prolonged release melatonin p.o./daily at bedtime	Aggregate of RBD incidents as captured in sleep diaries	Mild adverse effects (headaches, fatigue, morning sleepiness) in 4 subjects with melatonin and in 5 with placebo.	No	[143]
34 RBD patients	Prospective, open-label, randomized trial	4 weeks	Clonazepam 0.5 mg or prolonged-release melatonin 2 mg 30 minbefore bedtime	PSG, clinical global improvement-impression scale and sleepquestionnaire score	RBD symptom improvement tended to be better after clonazepam than after melatonin. Daytime sleepiness and insomnia symptoms were decreased by melatonin.	Clonazepam, but not PR melatonin, ↓ % RWA	[144]
12 iRBD patients	open-labeled trial		ramelteon 8 mg daily for 4 weeks	PSG	After ramelteon, a trend toward significant improvement occurred.	No statistically significant effect on RWA, RBD severity scale or other sleep parameters	[145]
35 patients, PD + sleep disorders; 24 with PRBD	open-labeled trial	12 weeks	ramelteon 8 mg daily,	RBD questionnaire	↓scores in the Japanese RBD questionnaire	No	[146]
59 year-old male RBD, 76 year-old female RBD, 1 PD, 1 multiple system atrophy + OSA	case series	2–3 years	ramelteon 8 mg monotherapy in 1 patient, +clonazepam 1 mg in 1 patient	PSG	Improvement in 2 patients, 1 with rebound after discontinuation.	↓ % RWA	[147]
3 iRBD patients	case review	6 months	agomelatine, 25–50 mg per night	PSG	3 (1 at 25 mg, 2 at 50 mg).	% of REM epochs with muscle atonia (atonic density) only slightly increased. In 2 cases the percentage of REM epochs with high tonic density decreased	[148]

IRBD: idiopathic rapid eye movement sleep behavior disorder; OSA: obstructive sleep apnea; PD: Parkinson disease; REM: rapid eye movement; RWA: REM sleep without atonia.

## Data Availability

Not applicable.

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
