# Peer review of "Melatonin as a Chronobiotic/Cytoprotective Agent in REM Sleep Behavior Disorder"

_brainsci, 2023, doi:10.3390/brainsci13050797_

Round 1

Reviewer 1 Report

Review of a manuscript “Melatonin as a Chronobiotic/Cytoprotective Agent in REM Sleep Behavior Disorder” by Daniel P. Cardinali and Arturo Garay submitted to “Brain Sciences.”

Rapid eye movement sleep without muscle atonia is considered as a parasomnia often termed REM sleep behavior disorder. RBD is a reliable prodromal marker of several neurodegenerative diseases collectively called α-synucleinopathies.  It is used as a biomarker for early identification of synucleinopathies. It is of paramount importance to identify Parkinson’s disease and other neurodegenerative diseases as soon as possible to timely begin neuroprotection measures and treatment. The authors discuss applications of melatonin in REM sleep behavior disorder, including a symptomatic treatment and as a possible disease-modifying therapy in various forms of synucleinopathies. This is an important area of biomedical research, and the data and discussion  presented in the manuscript will be interesting for the readers of the journal.

The following corrections and additions should be made.

Abstract

Lines 11-13. “RBD constitutes a robust prodromal marker of α-synucleinopathies and to date, the best biomarker available to predicts diseases like Parkinson disease, multiple system atrophy and dementia with Lewy bodies”. 1) The authors should rewrite this sentence in a more cautious way, since it may be understood as an ideal and reliable biomarker is ultimately found. 2) “the best biomarker available to predicts” should be corrected as “the best biomarker available to predict”.

Lines 16-17: ”Therefore patients with RBD are candidates for neuroprotection trials that delay or prevent the phenoconversion to a pathology with abnormal α-synuclein metabolism” . The sentence should be rewritten as “Therefore patients with RBD are candidates for neuroprotection trials that delay or prevent the phenoconversion to a pathology with abnormal α-synuclein modifications”. Also the term phenoconversion in this context may be confusing, because usually phenoconversion is considered as the mismatch between the individual's genotype-based prediction of drug metabolism and the true capacity to metabolize drugs. It will be beneficial if the authors replace this confusing term making the sentence it easier to read.  

Introduction

Lines 29-30:  After the sentence “Rapid-eye-movement (REM) sleep behavior disorder (RBD) is a REM sleep parasomnia that shows abnormal motor behaviors emerging during REM sleep in a context of diminished normal hypotonia [1]” the authors should add the following sentence and reference ” Various visual disturbances are associated with synucleinopathies” [reference Maurage CA et al., Retinal involvement in dementia with Lewy bodies: a clue to hallucinations? Ann Neurol. 2003 Oct;54(4):542-7. doi: 10.1002/ana.10730. PMID: 14520672.]

4. Basic Physiology of Melatonin Relevant to RBD

This heading may be corrected as Basic Physiology and Biochemistry of Melatonin Relevant to RBD

Line 321 “The pathological signs of PD are Lewy bodies and neurites, composed of amyloid aggregates of misfolded α-synuclein [117].” After this sentence the authors should add a reference on a recent relevant review: “Synucleins: New Data on Misfolding, Aggregation and Role in Diseases Biomedicines 10 (12), 2022 :3241. doi: 10.3390/biomedicines10123241. PMID: 36551997; PMCID: PMC9775291.

Concluding remarks

Lines 523-524: “Nonetheless, it would be wise for the pharmaceutical business to research the possibility of a profitable and medically effective combination of melatonin with specific medications”.

Please, be more specific about “effective combination of melatonin with specific medications” What combination might be used?

Overall, interesting manuscript reviewing important new findings.

Author Response

REVIEWER 1

Abstract

Lines 11-13. “RBD constitutes a robust prodromal marker of α-synucleinopathies and to date, the best biomarker available to predicts diseases like Parkinson disease, multiple system atrophy and dementia with Lewy bodies”. 1) The authors should rewrite this sentence in a more cautious way, since it may be understood as an ideal and reliable biomarker is ultimately found. 2) “the best biomarker available to predicts” should be corrected as “the best biomarker available to predict”.

Lines 16-17: ”Therefore patients with RBD are candidates for neuroprotection trials that delay or prevent the phenoconversion to a pathology with abnormal α-synuclein metabolism” . The sentence should be rewritten as “Therefore patients with RBD are candidates for neuroprotection trials that delay or prevent the phenoconversion to a pathology with abnormal α-synuclein modifications”. Also the term phenoconversion in this context may be confusing, because usually phenoconversion is considered as the mismatch between the individual's genotype-based prediction of drug metabolism and the true capacity to metabolize drugs. It will be beneficial if the authors replace this confusing term making the sentence it easier to read.

The text was changed accordingly (lines 10 to 27)

Introduction

Lines 29-30: After the sentence “Rapid-eye-movement (REM) sleep behavior disorder (RBD) is a REM sleep parasomnia that shows abnormal motor behaviors emerging during REM sleep in a

context of diminished normal hypotonia [1]” the authors should add the following sentence and reference ” Various visual disturbances are associated with synucleinopathies” [reference Maurage CA et al., Retinal involvement in dementia with Lewy bodies: a clue to hallucinations? Ann Neurol. 2003 Oct;54(4):542-7. doi: 10.1002/ana.10730. PMID: 14520672.]

The text was changed accordingly (lines 33 to 34)

  1. Basic Physiology of Melatonin Relevant to RBD

This heading may be corrected as Basic Physiology and Biochemistry of Melatonin Relevant to RBD

The heading was changed accordingly (line 188)

Line 321 “The pathological signs of PD are Lewy bodies and neurites, composed of amyloid aggregates of misfolded α-synuclein [117].” After this sentence the authors should add a reference on a recent relevant review: “Synucleins: New Data on Misfolding, Aggregation and Role in Diseases Biomedicines 10 (12), 2022 :3241. doi: 10.3390/biomedicines10123241. PMID: 36551997; PMCID: PMC9775291.

The text was changed accordingly (lines 311 to 312)

Concluding remarks

Lines 523-524: “Nonetheless, it would be wise for the pharmaceutical business to research the possibility of a profitable and medically effective combination of melatonin with specific medications”.

Please, be more specific about “effective combination of melatonin with specific medications” What combination might be used?

The text was changed accordingly (lines 503 to 504)

Reviewer 2 Report

This review article focused on RBD as a prodromal marker of alpha-synucleinopathies in PD as well as a few other diseases. In particular, the use of melatonin supplementation as an intervention to prevent alpha-synucleinopathies was a major aspect of the paper.

Overall, I think this review was well-done, on an important topic, and provided a great deal of interesting information on the above topics. I think it will be of interest to readers of the journal and adds to the literature on the topic. Thus, I think the paper should eventually be published and I only have a set of minor comments related to the writing, grammar, and organization of the paper. I believe the resolution of these issues will make this review better and more valuable. These minor issues are listed below.

 1.      The abstract needs to have a clear purpose or purposes more clearly stated.

2.      Section 2. This section is overall pretty good but perhaps could be broken down into a few subsections. Most importantly, there are many paragraphs in this section (and in other places in the review) that are very short such as in two sentences. Many of these are either slightly disjointed from the other adjacent paragraphs or could have been combined into adjacent paragraphs on similar topics. Sometimes a brief set of facts are mentioned and the transitions to other paragraphs are not great. Better topic and concluding sentences are needed in some instances. Section 4 also has many short paragraphs like this.

3.      Line 185, a specific direct reference is needed to support the statement that RBD is the best biomarker to predict alpha-synucleinopathies

4.      Figure 1 is too small and too blurry, it needs to be bigger and with better resolution.

5.      Table 1, it would be better if the heading of the first column did not have the “t” of the word concomitant hanging into the next line.

6.      Although well-written overall there are a some typos and grammatical errors of different kinds. A thorough proofreading is needed. Here are just some of examples:

a.       Line 16 there needs to be a comma after “Therefore”

b.      Line 66 too many spaces after “basis of”

c.       Line 94 too many spaces after “networks,”

d.      Line 110 there should be a space after “However”

e.       Line 111. Period in wrong place after “modulation”, delete it

f.        Line 167 space is needed after “RBD”

g.      Line 184 “Despite of these limitations” should read “Despite these limitations.

h.      Line 242, too many spaces after “promotes”

i.        Line 416 too many spaces after “of”

Decent amount of minor typos etc.

Author Response

REVIEWER 2

  1. The abstract needs to have a clear purpose or purposes more clearly stated.

The Abstract was changed accordingly (lines 24 to 25)

  1. Section 2. This section is overall pretty good but perhaps could be broken down into a few subsections. Most importantly, there are many paragraphs in this section (and in other places in the review) that are very short such as in two sentences. Many of these are either slightly disjointed from the other adjacent paragraphs or could have been combined into adjacent paragraphs on similar topics. Sometimes a brief set of facts are mentioned and the transitions to other paragraphs are not great. Better topic and concluding sentences are needed in some instances. Section 4 also has many short paragraphs like this.

Following this comment, the number of paragraphs in section 2 and 4 were reduced to 7 and 10, respectively.

  1. Line 185, a specific direct reference is needed to support the statement that RBD is the best biomarker to predict alpha-synucleinopathies

The text was changed accordingly (lines 184 to 185)

  1. Figure 1 is too small and too blurry, it needs to be bigger and with better resolution.

We are including the original ppt image for editorial purposes.

  1. Table 1, it would be better if the heading of the first column did not have the “t” of the word concomitant hanging into the next line.

We improve outline of Table 1 as much as we could.

  1. Although well-written overall there are a some typos and grammatical errors of different kinds. A thorough proofreading is needed. Here are just some of examples:
  2. Line 16 there needs to be a comma after “Therefore”
  3. Line 66 too many spaces after “basis of”
  4. Line 94 too many spaces after “networks,”
  5. Line 110 there should be a space after “However”
  6. Line 111. Period in wrong place after “modulation”, delete it
  7. Line 167 space is needed after “RBD”
  8. Line 184 “Despite of these limitations” should read “Despite these limitations.
  9. Line 242, too many spaces after “promotes”
  10. Line 416 too many spaces after “of

All typos were corrected.